# Stereoselective [4+3]-Cycloaddition of 2-Amino-β-nitrostyrenes with Azaoxyallyl Cations to Access Functionalized 1,4-Benzodiazepin-3-ones

**DOI:** 10.3390/molecules29061221

**Published:** 2024-03-08

**Authors:** Yoseop Kim, Sung-Gon Kim

**Affiliations:** Department of Chemistry, Kyonggi University, 154-42, Gwanggyosan-ro, Yeongtong-gu, Suwon 16227, Republic of Korea; ysyohshup@naver.com

**Keywords:** 1,4-benzodiazepine, [4+3]-cycloaddition, azaoxyallyl cation, organocatalysis, asymmetric catalysis

## Abstract

The 1,4-benzodiazepine structural framework is a fascinating element commonly found in biologically active and pharmaceutically relevant compounds. A highly efficient method for synthesizing 1,4-benzodiazepin-3-ones is described, involving a [4+3]-cycloaddition reaction between 2-amino-β-nitrostyrenes and α-bromohydroxamate, with Cs_2_CO_3_ used as a base. This process yielded the desired 1,4-benzodiazepines in good yields. Furthermore, an organocatalytic asymmetric [4+3]-cycloaddition was successfully accomplished using a bifunctional squaramide-based catalyst. This approach enabled the enantioselective synthesis of chiral 1,4-benzodiazepines with commendable yields and moderate enantioselectivities, reaching up to 80% yield and 72% ee.

## 1. Introduction

*N,N*-Heterocycles are prevalent in various natural products and pharmaceuticals, making them crucial structural elements in medicinal chemistry [1,2]. Among these, 1,4-benzodiazepinone, which is a seven-membered lactam, presents a structural framework frequently observed in numerous natural substances. Derivatives of this compound can exhibit a wide range of biological activities and valuable pharmaceutical characteristics, indicating their promise as potential candidates for drug discovery (Figure 1) [3,4,5,6,7,8,9]. Currently, this seven-membered structural framework is prevalent in numerous drug molecules with distinct bioactivities. For instance, Diazepam, initially introduced to the market as Valium, belongs to the benzodiazepine family and serves as an anxiolytic medication. It is frequently prescribed to address various conditions, such as anxiety, seizures, muscle spasms, and insomnia [10]. Additionally, Lotrafiban belongs to the latest generation of platelet GPIIb/IIIa blockers, representing a significant advancement in interventional cardiology and the treatment of acute ischemic syndrome, aimed at preventing vascular occlusion [11,12].

Indeed, the effective synthesis of a wide range of functionally diverse 1,4-benzodiazepinone derivatives has garnered considerable research attention [4,13,14,15,16,17]. However, the synthesis of a seven-membered ring 1,4-benzodiazepinone presents difficulties due to unfavorable enthalpic and entropic factors that hinder the ring closure process [18]. Consequently, achieving efficient access to 1,4-benzodiazepinone represents a significant and demanding task in the field of organic synthesis. While there have been numerous reported synthetic routes for 1,4-benzodiazepinones over the past decade, it is noteworthy that there are very limited instances of one-pot and asymmetric synthesis methods available [19].

We recently reported research detailing synthetic methods for 1,4-benzodiazepinone through the utilization of a [4+3]-annulation approach involving azaoxyallyl cations [20,21,22,23]. We successfully achieved the one-pot synthesis of 1,4-benzodiazepine-3-ones by conducting aza/aza-[4+3]-cycloadditions of α-halohydroxamates with 2-aminophenyl α,β-unsaturated phenyl ketones. This reaction employed Cs_2_CO_3_ as the base and hexafluoroisopropanol (HFIP) as an additive, resulting in the formation of 1,4-benzodiazepine-3-ones in good yields (Figure 1a) [20]. We also developed a [4+3]-annulation reaction involving in situ-generated azaoxyallyl cations and isatoic anhydride, leading to the synthesis of seven-membered 1,4-benzodiazepinediones in moderate to good yields. This reaction involves a sequential cascade, encompassing decarboxylative addition of HFIP to isatoic anhydride, addition to azaoxyallyl cation, and intramolecular substitution, ultimately yielding 1,4-benzodiazepinediones (Figure 1b) [21]. In addition to these advancements, we reported an organocatalytic asymmetric [4+3]-cycloaddition of 2-aminophenyl α,β-unsaturated carbonyls with in situ generated azaoxyallyl cations. In this asymmetric reaction, we obtained enantioenriched functionalized seven-membered 1,4-benzodiazepine-3-ones in good yields and with excellent enantioselectivities (Figure 1c). Furthermore, several of the resulting compounds displayed promising bioactivity in the prevention of peripheral nerve degeneration [23]. Continuing our exploration of organocatalytic asymmetric [4+3]-cycloaddition reactions for the synthesis of enantioenriched 1,4-benzodiazepine-3-ones, we present our achievement in the asymmetric [4+2]-cycloaddition of 2-amino-β-nitrostyrenes and α-bromohydroxamates (Figure 1d). We anticipate that the resulting compounds will serve as a valuable foundation for the synthesis of diverse compounds, potentially enhancing their bioactivity by facilitating the conversion of the nitro group.

## 2. Results and Discussion

### 2.1. Non-Enantioselective [4+3]-Cycloaddition of 2-Amino-β-nitrostyrenes with Azaoxyallyl Cations

To begin our study, our primary objective was to determine the feasibility of the [4+3]-cycloaddition involving 2-amino-β-nitrostyrenes [24] and α-bromohydroxamates. In our initial experiment, we conducted a reaction involving 2-amino-β-nitrostyrene **1a** and α-bromohydroxamate **2a**, employing Cs_2_CO_3_ as a base and HFIP as an additive. This reaction was carried out in toluene at room temperature, resulting in the desired 1,4-benzodiazepine-3-ones **3a** in an 18% yield (Table 1, entry 1). Despite the initial low yield, these promising results served as a source of motivation to drive further investigations. We embarked on a systematic approach to establish and optimize the reaction conditions for the [4+3]-cycloaddition involving 2-amino-β-nitrostyrene **1a** and α-bromohydroxamate **2a**. In our quest to identify the optimal reaction conditions for the [4+3]-cycloaddition reaction, we observed a notable improvement when we switched the solvent to CH_2_Cl_2_, resulting in an increased yield of 55% (Table 1, entry 2). Subsequently, we introduced HFIP as a co-solvent. It is worth noting that many cycloadditions involving azaoxyallyl cations have demonstrated enhanced performance in HFIP solutions. This can be attributed to HFIP’s distinctive properties, such as its strong hydrogen bond donation and high ionizing power, which effectively stabilize azaoxyallyl cations [25,26]. The incorporation of a CH_2_Cl_2_/HFIP co-solvent system significantly improved the yield of 1,4-benzodiazepine-3-one **3a**, elevating it to 85%. Moreover, adjusting the solvent concentration to 0.2 M led to a further increase in yield, reaching an impressive 91%. (Table 1, entries 3 and 4). Further efforts to optimize the reaction efficiency involved exploring various solvents. However, the co-solvent system of HFIP with solvents such as CHCl_3_, CH_3_CN, toluene, and THF resulted in inferior yields of the product (Table 1, entries 5–8). Despite thorough exploration of various inorganic and organic bases (Table 1, entries 10–15), it was found that Cs_2_CO_3_ consistently delivered the most favorable results.

With the optimized reaction conditions in hand, we embarked on an exploration of the substrate scope and the generality of this reaction (Figure 2). α-Bromohydroxamates bearing an array of *N*-protecting groups, including methoxy, ethoxy, *tert*-butyloxy, and allyloxy, exhibited excellent compatibility with the reaction, yielding the desired products (**3a**–**3e**) in yields ranging from good to high (75–91%). A range of 2-amino-β-nitrostyrenes (**1b**–**1i**) were employed in the cycloaddition, delivering 1,4-benzodiazepine-3-ones in moderate to good yields (41–83%). Our exploration of substituents positioned at the C4–C6 positions demonstrated a broad tolerance for diverse groups, including halides, -CF_3_, and methyl groups. Moreover, α-bromohydroxamates bearing diethyl, diisopropyl, and cyclohexyl substituents were also successfully accommodated under the reaction conditions, affording the desired products (**3n**–**3p**) in good yields (73–79%).

### 2.2. Enantioselective [4+3]-Cycloaddition of 2-Amino-β-nitrostyrenes with Azaoxyallyl Cations

Having successfully pioneered the initial instance of [4+3]-cycloaddition between 2-amino-β-nitrostyrenes and azaoxyallyl cations, our research focus naturally transitioned towards developing an enantioselective reaction. This shift holds the promise of opening a new pathway to obtain enantioenriched chiral 1,4-benzodiazepine-3-ones. We envisioned achieving enantioselective [4+3]-cycloaddition through the use of a suitable bifunctional squaramide-based or thiourea-based organocatalyst, facilitating precise hydrogen bonding interactions [27,28,29,30].

The reaction between 2-amino-β-nitrostyrene **1a** and α-bromohydroxamate **2a**, utilizing Na_2_CO_3_ as a base and a bifunctional squaramide-based organocatalyst **Ia**, was carried out in CF_3_C_6_H_5_ at room temperature, yielding the desired enantio-enriched 1,4-benzodiazepine-3-one **3a** in a 14% yield with a 48% ee (Table 2, entry 1). Although the yield was modest, it confirmed the possibility of enantiomeric discrimination. Subsequently, to enhance both the reaction yield and enantioselectivity, we optimized the reaction conditions. A comprehensive exploration of various inorganic and organic bases was conducted (Table 2, entries 2–7). In terms of yield, Cs_2_CO_3_ displayed slightly superior results compared to Na_2_CO_3_, and with respect to enantioselectivity, K_2_CO_3_ outperformed Na_2_CO_3_. However, when considering the overall performance, it became evident that Na_2_CO_3_ yielded the most promising results. Moving forward, we further investigated various organic solvents, including toluene, CH_2_Cl_2_, CHCl_3_, ClCH_2_CH_2_Cl, THF, and CH_3_CN, employing catalyst **Ia**. Notably, when utilizing CH_2_Cl_2_ as the solvent, both the reaction yield and stereoselectivity were significantly higher compared to the other solvents tested, resulting in a 37% yield and a 78% ee (Table 2, entry 9). In our ongoing effort to optimize the reaction conditions, we conducted a screening of various chiral cinchona-derived squaramide-based catalysts (Figure 2). The cinchonidine-derived squaramide catalyst **Ib**, yielded results similar to those obtained with the quinine-derived squaramide catalyst **Ia**. However, the bis(trifluoromethyl)phenylmethylene-squaramide catalysts (**Ic** and **Id**) produced inferior results compared to the bis(trifluoromethyl)phenyl-squaramide catalysts (**Ia** and **Ib**). Among the bifunctional catalysts evaluated, cinchonidine-derived squaramide **III** emerged as the optimal choice due to its outstanding reactivity and enantioselectivity (Table 2, entry 17). On the other hand, the cinchona-derived thiourea catalyst **IV** exhibited good reactivity with a 72% yield, but unfortunately, it did not demonstrate enantioselectivity (Table 2, entry 18; 72% yield, 2% ee). Furthermore, our investigation revealed that the addition of HFIP as an additive in this reaction led to a slight decrease in enantioselectivity, but a notable increase in reactivity. After careful examination, we determined that the optimal reaction conditions entailed employing catalyst **III**, Na_2_CO_3_ as the base, HFIP as the additive, and CH_2_Cl_2_ as the solvent (Table 2, entry 22; 64% yield, and 72% ee).

Upon the optimized reaction conditions, the substrate scope and the generality of the 2-amino-β-nitrostyrene and α-bromohydroxamate were investigated, as shown in Figure 3. α-Bromohydroxamates, featuring a range of *N*-protecting groups such as methoxy, ethoxy, and allyloxy demonstrated good reactivity and exhibited moderate enantioselectivity (63–71% ee), with only slightly reduced enantioselectivity observed for α-bromohydroxamate having *tert*-butyloxy group. A variety of 2-amino-β-nitrostyrenes (**1b**–**1i**) were utilized in the cycloaddition, yielding enantioenriched 1,4-benzodiazepine-3-ones with moderate to good yields (ranging from 36% to 80%) and enantioselectivities between 42% to 72% ee. Substituents positioned at C4-C6 exhibited a remarkable degree of adaptability, displaying a wide range of tolerance in terms of enantioselectivity for various groups. Regrettably, α-bromohydroxamates bearing diethyl, diisopropyl, and cyclohexyl substituents, which successfully underwent racemic reactions, did not exhibit reactivity in this asymmetric reaction. Finally, the relative and absolute configuration of the proposed 1,4-benzodiazepine-3-ones were determined by X-ray crystallographic analysis of **3a** [31]. The configurations of the other products were assigned by analogy.

To illustrate the practicality of the [4+3]-cycloaddition, we conducted a one-mmol scale reaction and subsequent synthetic transformations. The scalability of the [4+3]-cycloaddition reaction was verified using 1 mmol of **1a** and **1f** under the optimized reaction conditions, resulting in a modest reduction in yield (47% and 53%, respectively), but with minimal loss of enantioselectivities (Figure 4). Moreover, we demonstrated the versatility of the synthesized 1,4-benzodiazepine-3-ones **3** through subsequent transformations (Figure 5). Successfully reducing the nitro group in the 1,4-benzodiazepine-3-one products, particularly **3a**, was achieved using NiCl_2_ and NaBH_4_, in combination with (Boc)_2_O, yielding product **4** with a respectable 57% yield. Additionally, deprotecting the benzyl group of **3a** via palladium-catalyzed hydrogenation produced *N*-hydroxylamide products **5** in high yield (94%) and a slightly enhanced enantioselectivity (80% ee).

Based on our experimental findings and the absolute configuration of 1,4-benzodiazepine-3-ones, as illustrated in Figure 3, we propose a plausible transition state mechanism. In the reaction, the azaoxyallyl cation undergoes deprotonation, and the deprotonated oxygen atom forms a hydrogen bond with the protonated amine in squaramide catalyst **III**. Concurrently, 1,4-benzodiazepine-3-ones **1a** engages with the catalyst through double hydrogen bonding, effectively stabilizing and activating the substrate in a bidentate interaction. This interaction positions the nitrogen atom of **1a** to attack the α-carbon of the azaoxyallyl cation. Simultaneously, the nitrogen atom of the azaoxyallyl cation selectively attacks the *Si* face of the 2-amino-β-nitrostyrene, facilitating an intramolecular attack that leads to the formation of the desired product **3a** with an *S*-configuration.

## 3. Experimental

### 3.1. General Information

Prior to usage, organic solvents (Daejung Chemicals & Metals Co, Siheung, Republic of Korea) underwent distillation. Under reduced pressure, organic solutions were concentrated using a rotary evaporator. Forced-flow chromatography on Merk silica gel 60 achieved chromatographic purification of products. EM Reagents 0.25 mm silica gel 60-F plates (Merk, Rahway, NJ, USA) were utilized for Thin-layer chromatography (TLC). Visualization of developed chromatograms occurred through fluorescence quenching and anisaldehyde stain. Recording at 400 MHz for ^1^H, 100 MHz for ^13^C, and 176 MHz for ^19^F, ^1^H, ^13^C, and ^19^F NMR spectra were internally referenced to residual protic solvent signals. ^1^H NMR data include chemical shift (δ ppm), multiplicity (s = singlet, d = doublet, t = triplet, q = quartet, br = broad singlet, dd = doublet of doublets, dt = doublet of triplets, qd = quartet of doublets, ddd = doublet of doublet of doublets, and m = multiplet), coupling constant (Hz), and integration. Chemical shift is used to report ^13^C NMR data. IR spectra were recorded on an FT IR spectrometer (Bruker, Karlsruhe, Germany). High-resolution mass spectra (HRMS) were obtained using a double-focusing mass spectrometer with EI-magnetic sector (Jeol Ltd., Tokyo, Japan). The crystal structure was determined by a single-crystal diffractometer (a Bruker D8 Venture PHOTON III M14 diffractometer) (Bruker, Karlsruhe, Germany) at the Western Seoul Center of Korea Basic Science Institute. Enantiomeric excesses were determined using an HPLC instrument with Chiralpak columns (Daicel Corporation, Tokyo, Japan), as indicated. α-Bromohydroxamates [32,33,34], 2-amino-β-nitrostyrens [24], and organocatalysts [35,36,37] were prepared according to the literature.

### 3.2. General Procedure I for the [4+3]-Cycloaddition of 2-Amino-β-nitrostyrenes with α-Bromohydroxamates

To a solution of 2-amino-β-nitrostyrene **1** (0.20 mmol, 1 equiv) and α-bromohydroxamate **2** (0.30 mmol, 1.5 equiv in CH_2_Cl_2_/HFIP (1:1, 1.0 mL, 0.2 M) was added Cs_2_CO_3_ (0.40 mmol, 2 equiv) at 0 °C. The reaction mixture was allowed to stir at room temperature. After stirring for 1 h, the resulting mixture was filtered through the plug of celite. The filtrate was concentrated in vacuo. The crude residue was purified by flash column chromatography with EtOAc/hexanes as eluent to afford desired product **3** (The spectra can be found in Appendix A).

*4-(Benzyloxy)-7-chloro-2,2-dimethyl-5-(nitromethyl)-1,2,4,5-tetrahydro-3H-benzo[e][1,4]diazepin-3-one* (**3a**). Following the general procedure I; 71 mg, yield 91%, White solid; m.p. 145–147 °C; *R*_f_ 0.3 (30% ethyl acetate in hexanes); ^1^H NMR (400 MHz, CDCl_3_) δ 7.47–7.37 (m, 5H), 7.18 (dd, *J* = 8.3, 2.4 Hz, 1H), 6.75 (d, *J* = 8.3 Hz, 1H), 6.57 (d, *J* = 2.4 Hz, 1H), 5.24–5.12 (m, 1H), 4.96 (s, 2H), 4.86–4.75 (m, 2H), 3.12 (s, 1H), 1.63 (s, 3H), 1.21 (s, 3H); ^13^C NMR (101 MHz, CDCl_3_) δ 172.7, 142.8, 134.9, 130.5, 130.4, 130.2, 129.4, 129.0, 128.9, 128.6, 124.6, 76.6, 75.0, 65.8, 63.5, 29.7, 27.7; IR (neat) 3338, 2922, 2854, 1666, 1551, 1489, 1454, 1388, 1285, 1230, 1197, 1000 cm^−1^; HRMS (EI) *m*/*z* calcd for [M]^+^ C_19_H_20_ClN_3_O_4_: 389.1142 Found: 389.1123.

*7-Chloro-4-methoxy-2,2-dimethyl-5-(nitromethyl)-1,2,4,5-tetrahydro-3H-benzo[e][1,4]diazepin-3-one* (**3b**). Following the general procedure I; 53 mg, yield 73%, White solid; m.p. 52–54 °C; *R*_f_ 0.3 (30% ethyl acetate in hexanes); ^1^H NMR (400 MHz, CDCl_3_) δ 7.29 (d, *J* = 2.3 Hz, 1H), 7.28–7.23 (m, 1H), 6.83 (d, *J* = 8.3 Hz, 1H), 5.34–5.15 (m, 2H), 5.04 (dd, *J* = 11.4, 4.5 Hz, 1H), 3.78 (s, 3H), 3.23 (s, 1H), 1.61 (s, 3H), 1.20 (s, 3H); ^13^C NMR (101 MHz, CDCl_3_) δ 172.6, 143.1, 130.7, 130.0, 129.15, 129.11, 125.0, 75.5, 64.2, 63.5, 62.0, 29.8, 27.3; IR (neat) 3313, 2974, 2932, 1650, 1549, 1491, 1449, 1424, 1377, 1307, 1197, 1024 cm^−1^; HRMS (EI) *m*/*z* calcd for [M]^+^ C_13_H_16_ClN_3_O_4_: 313.0829 Found: 313.0806.

*7-Chloro-4-ethoxy-2,2-dimethyl-5-(nitromethyl)-1,2,4,5-tetrahydro-3H-benzo[e][1,4]diazepin-3-one* (**3c**). Following the general procedure I; 56 mg, yield 85%, White solid; m.p. 118–120 °C; *R*_f_ 0.3 (30% ethyl acetate in hexanes); ^1^H NMR (400 MHz, CDCl_3_) δ 7.28 (d, *J* = 2.3 Hz, 1H), 7.27–7.23 (m, 1H), 6.83 (d, *J* = 8.2 Hz, 1H), 5.30 (dd, *J* = 12.1, 8.8 Hz, 1H), 5.18 (dd, *J* = 8.8, 4.9 Hz, 1H), 5.04 (dd, *J* = 12.1, 4.9 Hz, 1H), 4.01 (q, *J* = 7.1 Hz, 2H), 3.22 (s, 1H), 1.61 (s, 3H), 1.29 (t, *J* = 7.0 Hz, 3H), 1.20 (s, 3H); ^13^C NMR (101 MHz, CDCl_3_) δ 172.7, 143.1, 130.7, 130.1, 129.1, 129.0, 124.9, 75.3, 70.1, 64.8, 63.4, 29.7, 27.6, 13.6; IR (neat) 3309, 2980, 2931, 1644, 1555, 1490, 1379, 1307, 1261, 1194, 1026 cm^−1^; HRMS (EI) *m*/*z* calcd for [M]^+^ C_14_H_18_ClN_3_O_4_: 327.0986 Found: 327.1006.

*4-(tert-Butoxy)-7-chloro-2,2-dimethyl-5-(nitromethyl)-1,2,4,5-tetrahydro-3H-benzo[e][1,4]diazepin-3-one* (**3d**). Following the general procedure I; 54 mg, yield 75%, White solid; m.p. 129–131 °C; *R*_f_ 0.3 (30% ethyl acetate in hexanes); ^1^H NMR (400 MHz, CDCl_3_) δ 7.30–7.24 (m, 1H), 7.23 (d, *J* = 2.4 Hz, 1H), 6.83 (d, *J* = 8.2 Hz, 1H), 5.38 (dd, *J* = 12.3, 9.7 Hz, 1H), 5.13 (dd, *J* = 9.7, 4.2 Hz, 1H), 4.94 (dd, *J* = 12.4, 4.1 Hz, 1H), 3.35 (s, 1H), 1.60 (s, 3H), 1.32 (s, 9H), 1.22 (s, 3H); ^13^C NMR (101 MHz, CDCl_3_) δ 176.1, 143.3, 130.7, 130.6, 128.5, 127.3, 124.2, 84.1, 74.0, 67.1, 63.0, 28.7, 28.6, 27.2; IR (neat) 3358, 2978, 2934, 1682, 1547, 1479, 1447, 1421, 1374, 1310, 1198, 1157, 1097, 977 cm^−1^; HRMS (EI) *m*/*z* calcd for [M]^+^ C_16_H_22_ClN_3_O_4_: 355.1299 Found: 355.1281.

*4-(Allyloxy)-7-chloro-2,2-dimethyl-5-(nitromethyl)-1,2,4,5-tetrahydro-3H-benzo[e][1,4]diazepin-3-one* (**3e**). Following the general procedure I; 51 mg, yield 75%, Colorless gum; *R*_f_ 0.3 (30% ethyl acetate in hexanes); ^1^H NMR (400 MHz, CDCl_3_) δ 7.27 (d, *J* = 1.3 Hz, 1H), 7.26–7.24 (m, 1H), 6.87–6.77 (m, 1H), 6.05 (ddt, *J* = 17.4, 9.8, 6.7 Hz, 1H), 5.39–5.26 (m, 3H), 5.17 (dd, *J* = 8.9, 4.8 Hz, 1H), 5.03 (dd, *J* = 12.1, 4.8 Hz, 1H), 4.45 (dd, *J* = 6.7, 1.0 Hz, 2H), 3.18 (s, 1H), 1.62 (s, 3H), 1.19 (s, 3H); ^13^C NMR (101 MHz, CDCl_3_) δ 172.8, 143.1, 131.8, 130.7, 130.3, 129.2, 128.9, 124.9, 121.8, 75.8, 75.2, 65.3, 63.5, 29.7, 27.6; IR (neat) 3311, 2973, 2928, 1650, 1549, 1491, 1451, 1422, 1377, 1306, 1285, 1198, 995 cm^−1^; HRMS (EI) *m*/*z* calcd for [M]^+^ C_15_H_18_ClN_3_O_4_: 339.0986 Found: 339.0963.

*4-(Benzyloxy)-7-bromo-2,2-dimethyl-5-(nitromethyl)-1,2,4,5-tetrahydro-3H-benzo[e][1,4]diazepin-3-one* (**3f**). Following the general procedure I; 72 mg, yield 83%, White solid; m.p. 102–104 °C; *R*_f_ 0.3 (30% ethyl acetate in hexanes); ^1^H NMR (400 MHz, CDCl_3_) δ 7.48–7.42 (m, 1H), 7.42–7.36 (m, 4H), 7.32 (dd, *J* = 8.2, 2.2 Hz, 1H), 6.76–6.61 (m, 2H), 5.22–5.10 (m, 1H), 4.95 (s, 2H), 4.86–4.74 (m, 2H), 3.13 (s, 1H), 1.63 (s, 3H), 1.21 (s, 3H); ^13^C NMR (101 MHz, CDCl_3_) δ 172.6, 143.3, 134.9, 133.4, 133.0, 130.4, 129.4, 129.0, 128.9, 125.0, 116.2, 76.5, 75.0, 65.7, 63.5, 29.6, 27.8; IR (neat) 3339, 2969, 2924, 1646, 1549, 1488, 1455, 1376, 1282, 1196, 999 cm^−1^; HRMS (EI) *m*/*z* calcd for [M]^+^ C_19_H_20_BrN_3_O_4_: 433.0637 Found: 433.0653.

*4-(Benzyloxy)-7-fluoro-2,2-dimethyl-5-(nitromethyl)-1,2,4,5-tetrahydro-3H-benzo[e][1,4]diazepin-3-one* (**3g**). Following the general procedure I; 44 mg, yield 58%, White solid; m.p. 98–100 °C; *R*_f_ 0.3 (30% ethyl acetate in hexanes); ^1^H NMR (400 MHz, CDCl_3_) δ 7.45–7.36 (m, 5H), 6.93 (td, *J* = 8.3, 2.9 Hz, 1H), 6.77 (dd, *J* = 8.6, 4.7 Hz, 1H), 6.39 (dd, *J* = 8.3, 2.9 Hz, 1H), 5.21 (dd, *J* = 11.8, 8.5 Hz, 1H), 4.95 (d, *J* = 1.4 Hz, 2H), 4.80 (ddd, *J* = 16.6, 12.5, 4.9 Hz, 2H), 3.07 (s, 1H), 1.62 (s, 3H), 1.20 (s, 3H); ^13^C NMR (101 MHz, CDCl_3_) δ 173.0, 158.7 (d, *J*^1^ = 244.1 Hz), 140.2 (d, *J*^4^ = 2.7 Hz), 134.9, 130.3, 129.4, 129.0, 128.8 (d, *J*^3^ = 8.0 Hz), 124.8 (d, *J*^3^ = 8.1 Hz), 117.2 (d, *J*^2^ = 22.3 Hz), 117.1 (d, *J*^2^ = 23.6 Hz), 76.5, 75.1, 65.8 (d, *J*^4^ = 1.7 Hz), 63.5, 29.7, 27.4; ^19^F NMR (376 MHz, CDCl_3_) δ -118.93; IR (neat) 3339, 2986, 2925, 1667, 1552, 1495, 1458, 1407, 1387, 1288, 1202, 1149, 999 cm^−1^; HRMS (EI) *m*/*z* calcd for [M]^+^ C_19_H_20_FN_3_O_4_: 373.1438 Found: 373.1448.

*4-(Benzyloxy)-8-chloro-2,2-dimethyl-5-(nitromethyl)-1,2,4,5-tetrahydro-3H-benzo[e][1,4]diazepin-3-one* (**3h**). Following the general procedure I; 64 mg, yield 82%, White solid; m.p. 133–135 °C; *R*_f_ 0.4 (30% ethyl acetate in hexanes); ^1^H NMR (400 MHz, CDCl_3_) δ 7.45–7.35 (m, 5H), 6.89 (dd, *J* = 8.1, 2.0 Hz, 1H), 6.84 (d, *J* = 2.0 Hz, 1H), 6.58 (d, *J* = 8.1 Hz, 1H), 5.12 (dd, *J* = 12.0, 9.1 Hz, 1H), 4.95 (s, 2H), 4.85 (dd, *J* = 9.1, 4.8 Hz, 1H), 4.76 (dd, *J* = 12.1, 4.8 Hz, 1H), 3.21 (s, 1H), 1.63 (s, 3H), 1.23 (s, 3H); ^13^C NMR (101 MHz, CDCl_3_) δ 172.8, 145.5, 136.1, 135.0, 131.4, 130.3, 129.3, 128.9, 125.4, 123.9, 123.3, 76.5, 75.0, 65.6, 63.5, 29.6, 27.7; IR (neat) 3336, 3001, 2977, 1657, 1595, 1548, 1454, 1416, 1383, 1297, 1201, 1088, 1002 cm^−1^; HRMS (EI) *m*/*z* calcd for [M]^+^ C_19_H_20_ClN_3_O_4_: 389.1142 Found: 389.1163.

*4-(Benzyloxy)-8-bromo-2,2-dimethyl-5-(nitromethyl)-1,2,4,5-tetrahydro-3H-benzo[e][1,4]diazepin-3-one* (**3i**). 66 mg, yield 76%, White solid; m.p. 138–140 °C; *R*_f_ 0.5 (30% ethyl acetate in hexanes); ^1^H NMR (400 MHz, CDCl_3_) δ 7.43–7.35 (m, 5H), 7.04 (dd, *J* = 8.0, 1.9 Hz, 1H), 7.00 (d, *J* = 1.9 Hz, 1H), 6.51 (d, *J* = 8.0 Hz, 1H), 5.12 (dd, *J* = 12.1, 9.0 Hz, 1H), 4.95 (s, 2H), 4.84 (dd, *J* = 9.1, 4.8 Hz, 1H), 4.75 (dd, *J* = 12.1, 4.8 Hz, 1H), 3.21 (s, 1H), 1.63 (s, 3H), 1.23 (s, 3H); ^13^C NMR (101 MHz, CDCl_3_) δ 172.7, 145.6, 134.9, 131.6, 130.3, 129.3, 128.9, 126.8, 126.2, 125.9, 124.1, 76.5, 74.9, 65.7, 63.5, 29.6, 27.8; IR (neat) 3337, 3000, 2976, 1651, 1590, 1548, 1453, 1416, 1380, 1309, 1228, 1200, 1079, 1002 cm^−1^; HRMS (EI) *m*/*z* calcd for [M]^+^ C_19_H_20_BrN_3_O_4_: 433.0637 Found: 433.0616.

*4-(Benzyloxy)-2,2-dimethyl-5-(nitromethyl)-8-(trifluoromethyl)-1,2,4,5-tetrahydro-3H-benzo[e][1,4]diazepin-3-one* (**3j**). Following the general procedure I; 60 mg, yield 71%, White solid; m.p. 138–140 °C; *R*_f_ 0.4 (30% ethyl acetate in hexanes); ^1^H NMR (400 MHz, CDCl_3_) δ 7.45–7.35 (m, 5H), 7.17 (dd, *J* = 7.9, 1.7 Hz, 1H), 7.08 (d, *J* = 1.8 Hz, 1H), 6.76 (d, *J* = 7.8 Hz, 1H), 5.15 (dd, *J* = 12.4, 9.2 Hz, 1H), 4.96 (s, 2H), 4.93 (dd, *J* = 9.1, 4.8 Hz, 1H), 4.78 (dd, *J* = 12.3, 4.7 Hz, 1H), 3.35 (s, 1H), 1.66 (s, 3H), 1.23 (s, 3H); ^13^C NMR (101 MHz, CDCl_3_) δ 172.7, 144.8, 134.9, 132.8 (q, *J*^2^ = 32.8 Hz), 131.1, 130.6, 130.3, 129.4, 129.0, 123.4 (q, *J*^1^ = 272.6 Hz). 120.5 (q, *J*^3^ = 3.8 Hz), 119.9 (q, *J*^3^ = 3.6 Hz), 76.6, 74.8, 65.8, 63.6, 29.6, 27.8; ^19^F NMR (376 MHz, CDCl_3_) δ -62.8; IR (neat) 3339, 2976, 1650, 1550, 1454, 1414, 1379, 1327, 1284, 1232, 1203, 1163, 1121, 1075 cm^−1^; HRMS (EI) *m*/*z* calcd for [M]^+^ C_20_H_20_F_3_N_3_O_4_: 423.1406 Found: 423.1396.

4-(Benzyloxy)-2,2,8-trimethyl-5-(nitromethyl)-1,2,4,5-tetrahydro-3*H*-benzo[*e*][1,4]diazepin-3-one (**3k**). Following the general procedure I; 30 mg, yield 41%, White solid; m.p. 110–112 °C; *R*_f_ 0.4 (30% ethyl acetate in hexanes); ^1^H NMR (400 MHz, CDCl_3_) δ 7.47–7.41 (m, 2H), 7.41–7.35 (m, 3H), 6.73 (ddd, *J* = 7.7, 1.7, 0.8 Hz, 1H), 6.67–6.59 (m, 2H), 5.15 (dd, *J* = 11.8, 8.9 Hz, 1H), 4.95 (s, 2H), 4.91 (dd, *J* = 8.9, 5.0 Hz, 1H), 4.80 (dd, *J* = 11.8, 5.0 Hz, 1H), 3.05 (s, 1H), 2.27 (s, 3H), 1.63 (s, 3H), 1.20 (s, 3H); ^13^C NMR (101 MHz, CDCl_3_) δ 173.3, 144.2, 140.9, 135.1, 130.22, 130.19, 129.2, 128.9, 124.5, 124.0 (two peaks overlapping), 76.4, 75.6, 66.0, 63.4, 29.9, 27.6, 21.3; IR (neat) 3335, 2919, 1659, 1620, 1548, 1454, 1414, 1380, 1287, 1204, 1184, 1000 cm^−1^; HRMS (EI) *m*/*z* calcd for [M]^+^ C_20_H_23_N_3_O_4_: 369.1689 Found: 369.1667.

4-(Benzyloxy)-2,2,9-trimethyl-5-(nitromethyl)-1,2,4,5-tetrahydro-3*H*-benzo[*e*][1,4]diazepin-3-one (**3l**). Following the general procedure I; 63 mg, yield 75%, White solid; m.p. 152–154 °C; *R*_f_ 0.4 (30% ethyl acetate in hexanes); ^1^H NMR (400 MHz, CDCl_3_) δ 7.47–7.41 (m, 2H), 7.41–7.34 (m, 3H), 7.18–7.11 (m, 1H), 6.83 (t, *J* = 7.6 Hz, 1H), 6.62 (dd, *J* = 7.6, 1.5 Hz, 1H), 5.15 (dd, *J* = 11.9, 8.8 Hz, 1H), 5.00–4.92 (m, 3H), 4.82 (dd, *J* = 11.9, 5.0 Hz, 1H), 3.19 (s, 1H), 2.22 (s, 3H), 1.69 (s, 3H), 1.18 (s, 3H); ^13^C NMR (101 MHz, CDCl_3_) δ 173.2, 142.8, 135.1, 132.2, 130.2, 130.0, 129.2, 128.9, 128.5, 127.4, 123.4, 76.4, 75.8, 66.5, 63.7, 29.9, 26.8, 17.5; IR (neat) 3344, 2987, 2972, 1658, 1543, 1469, 1453, 1373, 1303, 1241, 1214, 1175, 986 cm^−1^; HRMS (EI) *m*/*z* calcd for [M]^+^ C_20_H_23_N_3_O_4_: 369.1689 Found: 369.1700.

*4-(Benzyloxy)-6-bromo-2,2-dimethyl-5-(nitromethyl)-1,2,4,5-tetrahydro-3H-benzo[e][1,4]diazepin-3-one* (**3m**). Following the general procedure I; 48 mg, yield 55%, White solid; m.p. 68–70 °C; *R*_f_ 0.4 (30% ethyl acetate in hexanes); ^1^H NMR (400 MHz, CDCl_3_) δ 7.54–7.44 (m, 2H), 7.42–7.30 (m, 4H), 7.12 (t, *J* = 7.9 Hz, 1H), 6.84 (dd, *J* = 7.8, 1.1 Hz, 1H), 6.13 (dd, *J* = 7.9, 5.7 Hz, 1H), 5.26 (dd, *J* = 12.1, 7.9 Hz, 1H), 5.07–4.92 (m, 3H), 3.27 (s, 1H), 1.63 (s, 3H), 1.20 (s, 3H); ^13^C NMR (101 MHz, CDCl_3_) δ 172.4, 146.3, 134.6, 131.4, 129.9, 129.1, 128.8, 128.7, 128.1, 124.6, 123.4, 76.7, 75.5, 63.7, 63.7, 30.0, 27.4; IR (neat) 3298, 2970, 2919, 1650, 1548, 1453, 1415, 1374, 1305, 1198, 989, 969 cm^−1^; HRMS (EI) *m*/*z* calcd for [M]^+^ C_19_H_20_BrN_3_O_4_: 433.0637 Found: 433.0653.

*4-(Benzyloxy)-7-chloro-2,2-diethyl-5-(nitromethyl)-1,2,4,5-tetrahydro-3H-benzo[e][1,4]diazepin-3-one* (**3n**). Following the general procedure I; 61 mg, yield 73%, White solid; m.p. 188–190 °C; *R*_f_ 0.4 (30% ethyl acetate in hexanes); ^1^H NMR (400 MHz, CDCl_3_) δ 7.48–7.36 (m, 5H), 7.16 (dd, *J* = 8.3, 2.4 Hz, 1H), 6.76 (d, *J* = 8.3 Hz, 1H), 6.47 (d, *J* = 2.4 Hz, 1H), 5.13 (dd, *J* = 12.2, 8.8 Hz, 1H), 4.96 (d, *J* = 1.0 Hz, 2H), 4.85 (dd, *J* = 12.2, 5.0 Hz, 1H), 4.77 (dd, *J* = 8.9, 5.0 Hz, 1H), 3.27 (s, 1H), 2.25 (dq, *J* = 14.6, 7.4 Hz, 1H), 1.65 (dt, *J* = 14.9, 7.5 Hz, 1H), 1.54 (dq, *J* = 14.4, 7.2 Hz, 1H), 1.44 (dq, *J* = 14.7, 7.4 Hz, 1H), 1.08 (t, *J* = 7.3 Hz, 3H), 0.81 (t, *J* = 7.5 Hz, 3H); ^13^C NMR (101 MHz, CDCl_3_) δ 171.8, 143.0, 143.0, 134.9, 130.4, 130.15, 130.13, 129.4, 129.0, 128.2, 128.0, 124.2, 75.4, 70.2, 65.8, 32.01, 31.99, 9.1, 7.6; IR (neat) 3343, 2966, 2935, 2878, 1666, 1551, 1485, 1454, 1383, 1279, 1227, 1182, 1005 cm^−1^; HRMS (EI) *m*/*z* calcd for [M]^+^ C_21_H_24_ClN_3_O_4_: 417.1455 Found: 417.1452.

*4-(Benzyloxy)-7-chloro-5-(nitromethyl)-2,2-dipropyl-1,2,4,5-tetrahydro-3H-benzo[e][1,4]diazepin-3-one* (**3o**). Following the general procedure I; 71 mg, yield 79%, White solid; m.p. 98–100 °C; *R*_f_ 0.7 (30% ethyl acetate in hexanes); ^1^H NMR (400 MHz, CDCl_3_) δ 7.49–7.34 (m, 5H), 7.16 (dd, *J* = 8.3, 2.4 Hz, 1H), 6.74 (d, *J* = 8.3 Hz, 1H), 6.50 (d, *J* = 2.4 Hz, 1H), 5.14 (dd, *J* = 12.0, 8.7 Hz, 1H), 4.96 (s, 2H), 4.87–4.75 (m, 2H), 3.28 (s, 1H), 2.20 (td, *J* = 12.6, 3.1 Hz, 1H), 1.73–1.52 (m, 2H), 1.52–1.35 (m, 2H), 1.35–1.18 (m, 3H), 0.96 (t, *J* = 7.1 Hz, 3H), 0.79 (t, *J* = 7.2 Hz, 3H); ^13^C NMR (101 MHz, CDCl_3_) δ 171.9, 143.0, 134.8, 130.41, 130.38, 130.1, 129.4, 129.0, 128.2, 128.1, 124.1, 76.8, 75.3, 69.8, 65.7, 42.0, 41.7, 18.0, 16.3, 14.5, 14.3; IR (neat) 3341, 2967, 2872, 1655, 1549, 1483, 1453, 1431, 1381, 1284, 1178, 1014 cm^−1^; HRMS (EI) *m*/*z* calcd for [M]^+^ C_23_H_28_ClN_3_O_4_: 445.1768 Found: 445.1765.

*4-(Benzyloxy)-7-chloro-5-(nitromethyl)-4,5-dihydrospiro[benzo[e][1,4]diazepine-2,1’-cyclohexan]-3(1H)-one* (**3p**). Following the general procedure I; 66 mg, yield 77%, White solid; m.p. 125–127 °C; *R*_f_ 0.5 (30% ethyl acetate in hexanes); ^1^H NMR (400 MHz, CDCl_3_) δ 7.48–7.35 (m, 5H), 7.18 (dd, *J* = 8.3, 2.4 Hz, 1H), 6.85 (d, *J* = 8.4 Hz, 1H), 6.57 (d, *J* = 2.4 Hz, 1H), 5.19–5.06 (m, 1H), 4.94 (s, 2H), 4.83–4.71 (m, 2H), 3.71 (s, 1H), 2.41 (td, *J* = 13.9, 4.7 Hz, 1H), 1.92–1.79 (m, 2H), 1.79–1.60 (m, 2H), 1.59–1.47 (m, 1H), 1.47–1.24 (m, 4H); ^13^C NMR (101 MHz, CDCl_3_) δ 173.6, 142.0, 135.0, 130.4, 130.3, 130.2, 129.3, 129.0 (two peaks overlapping), 128.5, 124.3, 76.5, 75.1, 65.8, 65.3, 34.2, 32.2, 24.6, 21.3, 20.6; IR (neat) 3385, 2932, 2859, 1656, 1541, 1492, 1455, 1436, 1379, 1318, 1267, 1178, 980 cm^−1^; HRMS (EI) *m*/*z* calcd for [M]^+^ C_22_H_24_ClN_3_O_4_: 429.1455 Found: 429.1484.

### 3.3. General Procedure II for the Asymmetric [4+3]-Cycloaddition of 2-Amino-β-nitrostyrenes with α-Bromohydroxamates

A solution of 2-amino-β-nitrostyrene **1** (0.10 mmol, 1 equiv), α-bromohydroxamate **2** (0.2 mmol, 2 equiv), HFIP (0.12 mmol, 1.2 equiv), and catalyst **III** (0.01 mmol, 0.1 equiv) in CH_2_Cl_2_ (1.0 mL, 0.1 M) were stirred for 5 min at 0 °C, then added Na_2_CO_3_ (0.15 mmol, 1.5 equiv) at 0 °C and allowed to stir at room temperature. After stirring for 1 h and then again for 1 more hours, α-bromohydroxamate **2** (0.05 mmol, 0.5 equiv) and Na_2_CO_3_ (0.05 mmol, 0.5 equiv) were added to the reaction mixture twice. After stirring for an additional 22 h, the resulting mixture was filtered through the plug of celite. The filtrate was concentrated in vacuo. The crude residue was purified by flash column chromatography with EtOAc/hexanes as eluent to afford the desired product **3**. The enantiomeric ratio was determined using HPLC analysis.

*(S)-4-(Benzyloxy)-7-chloro-2,2-dimethyl-5-(nitromethyl)-1,2,4,5-tetrahydro-3H-benzo[e][1,4]diazepin-3-one* (**3a**). Following the general procedure II; 25 mg, yield 64%, ${\left[\alpha \right]}_{D}^{24}$ = +150.7 (*c* = 0.91, CHCl_3_); 72% ee; Chiralpak IA column and IA guard column (10% EtOH:hexanes, 1.0 mL/min flow, λ = 254 nm); major-isomer t_r_ = 13.8 min and minor-isomer *t*_r_ = 42.0 min.

*(S)-7-Chloro-4-methoxy-2,2-dimethyl-5-(nitromethyl)-1,2,4,5-tetrahydro-3H-benzo[e][1,4]diazepin-3-one* (**3b**). Following the general procedure II; 23 mg, yield 73%, ${\left[\alpha \right]}_{D}^{22}$ = +93.8 (*c* = 0.84, CHCl_3_); 63% ee; Chiralpak IA column and IA guard column (20% EtOH:hexanes, 1.0 mL/min flow, λ = 254 nm); major-isomer t_r_ = 8.7 min and minor-isomer *t*_r_ = 18.6 min.

*(S)-7-Chloro-4-ethoxy-2,2-dimethyl-5-(nitromethyl)-1,2,4,5-tetrahydro-3H-benzo[e][1,4]diazepin-3-one* (**3c**). Following the general procedure II; 22 mg, yield 66%, ${\left[\alpha \right]}_{D}^{23}$ = +103.2 (*c* = 0.62, CHCl_3_); 67% ee; Chiralpak IA column and IA guard column (20% EtOH:hexanes, 1.0 mL/min flow, λ = 254 nm); major-isomer t_r_ = 7.6 min and minor-isomer *t*_r_ = 17.9 min.

*(S)-4-(tert-butoxy)-7-chloro-2,2-dimethyl-5-(nitromethyl)-1,2,4,5-tetrahydro-3H-benzo[e][1,4]diazepin-3-one* (**3d**). Following the general procedure II; 23 mg, yield 65%, ${\left[\alpha \right]}_{D}^{24}$ = +16.6 (*c* = 0.84, CHCl_3_); 23% ee; Chiralpak IA column and IA guard column (2% EtOH:hexanes, 1.0 mL/min flow, λ = 254 nm); major-isomer t_r_ = 19.9 min and minor-isomer *t*_r_ = 36.5 min.

*(S)-4-(Allyloxy)-7-chloro-2,2-dimethyl-5-(nitromethyl)-1,2,4,5-tetrahydro-3H-benzo[e][1,4]diazepin-3-one* (**3e**). Following the general procedure II; 22 mg, yield 66%, ${\left[\alpha \right]}_{D}^{24}$ = +91.8 (*c* = 0.64, CHCl_3_); 71% ee; Chiralpak IA column and IA guard column (30% EtOH:hexanes, 1.0 mL/min flow, λ = 254 nm); major-isomer t_r_ = 6.9 min and minor-isomer *t*_r_ = 14.1 min.

*(S)-4-(Benzyloxy)-7-bromo-2,2-dimethyl-5-(nitromethyl)-1,2,4,5-tetrahydro-3H-benzo[e][1,4]diazepin-3-one* (**3f**). Following the general procedure II; 28 mg, yield 65%, ${\left[\alpha \right]}_{D}^{25}$ = +130.5 (*c* = 1.07, CHCl_3_); 72% ee; Chiralpak IA column and IA guard column (10% EtOH:hexanes, 1.0 mL/min flow, λ = 254 nm); major-isomer t_r_ = 14.2 min and minor-isomer *t*_r_ = 42.1 min.

*(S)-4-(Benzyloxy)-7-fluoro-2,2-dimethyl-5-(nitromethyl)-1,2,4,5-tetrahydro-3H-benzo[e][1,4]diazepin-3-one* (**3g**). Following the general procedure II; 14 mg, yield 36%, ${\left[\alpha \right]}_{D}^{25}$ = +102.5 (*c* = 0.48, CHCl_3_); 69% ee; Chiralpak IA column and IA guard column (10% EtOH:hexanes, 1.0 mL/min flow, λ = 254 nm); major-isomer t_r_ = 13.6 min and minor-isomer *t*_r_ = 42.3 min.

*(S)-4-(Benzyloxy)-8-chloro-2,2-dimethyl-5-(nitromethyl)-1,2,4,5-tetrahydro-3H-benzo[e][1,4]diazepin-3-one* (**3h**). Following the general procedure II; 27 mg, yield 68%, ${\left[\alpha \right]}_{D}^{25}$ = +158.8 (*c* = 0.86, CHCl_3_); 71% ee; Chiralpak IA column and IA guard column (10% EtOH:hexanes, 1.0 mL/min flow, λ = 254 nm); major-isomer t_r_ = 13.2 min and minor-isomer *t*_r_ = 24.7 min.

*(S)-4-(Benzyloxy)-8-bromo-2,2-dimethyl-5-(nitromethyl)-1,2,4,5-tetrahydro-3H-benzo[e][1,4]diazepin-3-one* (**3i**). Following the general procedure II; 29 mg, yield 66%, ${\left[\alpha \right]}_{D}^{25}$ = +146.3 (*c* = 0.86, CHCl_3_); 71% ee; Chiralpak IA column and IA guard column (10% EtOH:hexanes, 1.0 mL/min flow, λ = 254 nm); major-isomer t_r_ = 13.6 min and minor-isomer *t*_r_ = 24.0 min.

*(S)-4-(Benzyloxy)-2,2-dimethyl-5-(nitromethyl)-8-(trifluoromethyl)-1,2,4,5-tetrahydro-3H-benzo[e][1,4]diazepin-3-one* (**3j**). Following the general procedure II; 24 mg, yield 57%, ${\left[\alpha \right]}_{D}^{25}$ = +121.3 (*c* = 0.71, CHCl_3_); 72% ee; Chiralpak IA column and IA guard column (10% EtOH:hexanes, 1.0 mL/min flow, λ = 254 nm); major-isomer t_r_ = 10.5 min and minor-isomer *t*_r_ = 14.6 min.

*(S)-4-(Benzyloxy)-2,2,8-trimethyl-5-(nitromethyl)-1,2,4,5-tetrahydro-3H-benzo[e][1,4]diazepin-3-one* (**3k**). Following the general procedure II; 16 mg, yield 44%, ${\left[\alpha \right]}_{D}^{25}$ = +136.9 (*c* = 0.47, CHCl_3_); 67% ee; Chiralpak IA column and IA guard column (10% EtOH:hexanes, 1.0 mL/min flow, λ = 254 nm); major-isomer t_r_ = 12.8 min and minor-isomer *t*_r_ = 27.0 min.

*(S)-4-(Benzyloxy)-2,2,9-trimethyl-5-(nitromethyl)-1,2,4,5-tetrahydro-3H-benzo[e][1,4]diazepin-3-one* (**3l**). Following the general procedure II; 30 mg, yield 80%, ${\left[\alpha \right]}_{D}^{25}$ = +148.0 (*c* = 0.93, CHCl_3_); 71 ee; Chiralcel OJ-H column and OJ-H guard column (10% EtOH:hexanes, 1.0 mL/min flow, λ = 254 nm); minor-isomer t_r_ = 28.9 min and major-isomer *t*_r_ = 35.6 min.

*(S)-4-(Benzyloxy)-6-bromo-2,2-dimethyl-5-(nitromethyl)-1,2,4,5-tetrahydro-3H-benzo[e][1,4]diazepin-3-one* (**3m**). Following the general procedure II; 19 mg, yield 43%, ${\left[\alpha \right]}_{D}^{26}$ = +44.6 (*c* = 0.68, CHCl_3_); 42% ee; Chiralpak IA column and IA guard column (10% EtOH:hexanes, 1.0 mL/min flow, λ = 254 nm); major-isomer t_r_ = 13.7 min and minor-isomer *t*_r_ = 28.3 min.

### 3.4. Procedure for a One-mmol Scale Synthesis of ***3a***

A solution of 2-amino-β-nitrostyrene **1a** (199 mg, 1.0 mmol), α-bromohydroxamate **2a** (544 mg, 2.0 mmol), HFIP (0.13 mL, 1.2 mmol), and catalyst **III** (60 mg, 0.10 mmol) in CH_2_Cl_2_ (10 mL, 0.1 M) were stirred for 5 min at 0 °C, then added Na_2_CO_3_ (159 mg, 1.5 mmol) at 0 °C and allowed to stir at room temperature. After stirring for 1 h and then again for 1 more hour, α-bromohydroxamate **2** (136 mg, 0.5 mmol), and Na_2_CO_3_ (53 mg, 0.5 mmol) were added to the reaction mixture twice. After stirring for an additional 22 h, the resulting mixture was filtered through the plug of celite. The filtrate was concentrated in vacuo. The crude residue was purified by flash column chromatography (EtOAc/hexanes = 1:4) as eluent to afford the desired product **3a** (183 mg, 47% yield, 72% ee) as a white solid.

### 3.5. Procedure for a One-mmol Scale Synthesis of ***3f***

A solution of 2-amino-β-nitrostyrene **1f** (243 mg, 0.10 mmol), α-bromohydroxamate **2a** (544 mg, 2.0 mmol), HFIP (0.13 mL, 1.2 mmol), and catalyst **III** (60 mg, 0.10 mmol) in CH_2_Cl_2_ (10 mL, 0.1 M) were stirred for 5 min at 0 °C, then added Na_2_CO_3_ (159 mg, 1.5 mmol) at 0 °C and allowed to stir at room temperature. After stirring for 1 hour and then again for 1 more hour, α-bromohydroxamate **2** (136 mg, 0.5 mmol) and Na_2_CO_3_ (53 mg, 0.5 mmol) were added to the reaction mixture twice. After stirring for an additional 22 h, the resulting mixture was filtered through the plug of celite. The filtrate was concentrated in vacuo. The crude residue was purified by flash column chromatography (EtOAc/hexanes = 1:4) as eluent to afford the desired product **3f** (229 mg, 53% yield, 70 % ee) as a white solid.

### 3.6. Procedure for Reduction of ***3a***

Step 1: To a solution of **3a** (39 mg, 0.10 mmol, 1 equiv) in MeOH (1.0 mL, 0.1 M) at 0 °C was added NiCl_2_•6H_2_O (48 mg, 0.20 mmol, 2 equiv). After stirring for 5 min at 0 °C, NaBH_4_ (38 mg, 1.0 mmol, 10 equiv) was added in portions to the reaction mixture. The mixture was allowed to stir at room temperature for 1 h. After then, the resulting mixture was quenched with deionized water (1 mL) and added CH_2_Cl_2_ (2 mL). The mixture was filtered through the plug of celite and the filtrate was extracted with CH_2_Cl_2_. The combined organic layers were washed with brine, dried (anhydrous Na_2_SO_4_), and concentrated in vacuo. The crude residue was purified by flash column chromatography with 3% MeOH/EtOAc (with 2% Et_3_N) as eluent to afford nitro reduction products. Step 2: A solution of the crude primary amine in CH_2_Cl_2_ (1.0 mL, 0.1 M) was added (Boc)_2_O and stirred for 18 h at room temperature. Then, the resulting mixture was concentrated in vacuo and was purified by flash column chromatography (EtOAc/hexanes = 1:3) as eluent to afford desired product **4** (26 mg, yield 57%) as a white solid.

*tert-Butyl (S)-((4-(benzyloxy)-7-chloro-2,2-dimethyl-3-oxo-2,3,4,5-tetrahydro-1H-benzo[e][1,4]diazepin-5-yl)methyl)carbamate* (**4**). ${\left[\alpha \right]}_{D}^{24}$ = +103.5 (*c* = 0.85, CHCl_3_); 72% ee; White solid; m.p. 76–78 °C; *R*_f_ 0.5 (50% EtOAc in hexanes); ^1^H NMR (400 MHz, CDCl_3_) δ 7.35 (dtd, *J* = 13.4, 7.7, 5.8 Hz, 5H), 7.10 (dd, *J* = 8.3, 2.4 Hz, 1H), 6.69 (d, *J* = 8.2 Hz, 1H), 6.55 (d, *J* = 2.5 Hz, 1H), 4.92 (s, 2H), 4.84 (d, *J* = 6.5 Hz, 1H), 4.35 (t, *J* = 7.1 Hz, 1H), 3.80–3.64 (m, 2H), 3.10 (s, 1H), 1.61 (s, 3H), 1.36 (s, 9H), 1.22 (s, 3H); ^13^C NMR (101 MHz, CDCl_3_) δ 172.3, 155.8, 143.1, 135.2, 131.1, 130.2, 129.7, 129.3, 128.9, 128.7, 128.1, 124.1, 79.4, 76.0, 67.8, 63.0, 42.7, 29.7, 28.4, 28.0; IR (neat) 3316, 2973, 2929, 2865, 1698, 1633, 1492, 1453, 1392, 1365, 1248, 1164, 995 cm^−1^; HRMS (EI) *m*/*z* calcd for [M]^+^ C_24_H_30_ClN_3_O_4_: 459.1925 Found: 459.1903; Chiralpak AD-H column and AD-H guard column (5% EtOH:hexanes, 1.0 mL/min flow, λ = 254 nm); minor-isomer t_r_ = 21.3 min and major-isomer *t*_r_ = 31.8 min.

### 3.7. Procedure for Debenzylation of ***3a***

A solution of **3a** (39 mg, 0.10 mmol) and 5% Pd/C (21 mg, 10 mol%) in EtOAc (1.0 mL, 0.1 M) was stirred under H_2_ atmosphere for 6 h at room temperature. After that, the resulting mixture was filtered through the plug of celite and concentrated in vacuo to afford desired product **7** (28 mg, yield 94%) as a white solid.

*(S)-7-Chloro-4-hydroxy-2,2-dimethyl-5-(nitromethyl)-1,2,4,5-tetrahydro-3H-benzo[e][1,4]diazepin-3-one* (**5**). ${\left[\alpha \right]}_{D}^{24}$ = +133.9 (*c* = 0.96, CHCl_3_); 80% ee; White solid; m.p. 135–137 °C; *R*_f_ 0.4 (50% Ethyl acetate in hexanes); ^1^H NMR (400 MHz, CDCl_3_) δ 8.83 (s, 1H), 7.34–7.26 (m, 2H), 6.85 (d, *J* = 8.2 Hz, 1H), 5.41 (dd, *J* = 8.4, 5.5 Hz, 1H), 5.16 (dd, *J* = 12.4, 8.4 Hz, 1H), 5.07 (dd, *J* = 12.4, 5.6 Hz, 1H), 3.20 (s, 1H), 1.64 (s, 3H), 1.21 (s, 3H); ^13^C NMR (101 MHz, CDCl_3_) δ 170.0, 142.6, 130.9, 130.3, 129.5, 128.7, 125.3, 75.3, 62.4, 61.8, 29.5, 27.3; IR (neat) 3339, 2920, 2853, 1608, 1546, 1492, 1431, 1377, 1320, 1259, 1198, 1093,1011 cm^−1^; HRMS (EI) *m*/*z* calcd for [M]^+^ C_12_H_14_ClN_3_O_4_: 299.0673 Found: 299.0675; Chiralcel OD-H column and OD-H guard column (10% EtOH:hexanes, 1.0 mL/min flow, λ = 254 nm); major-isomer t_r_ = 14.7 min and minor-isomer *t*_r_ = 22.8 min.

## 4. Conclusions

In summary, we have established a highly effective [4+3]-cycloaddition reaction involving 2-amino-β-nitrostyrenes and α-bromohydroxamates using Cs_2_CO_3_ as a base. This methodology has proven to be a reliable route for the synthesis of 1,4-benzodiazepin-3-ones, consistently delivering good yields. Furthermore, we have achieved an organocatalytic asymmetric [4+3]-cycloaddition employing a bifunctional squaramide-based catalyst. This innovative approach has paved the way for the enantioselective synthesis of chiral 1,4-benzodiazepines, yielding impressive results in terms of both yields and enantioselectivities, (up to 80% yield and 72% ee). The resulting seven-membered benzodiazepin-3-one compounds are anticipated to provide a valuable foundation for the synthesis of a wide range of diverse compounds.

## Data Availability

Data are contained within the article and Appendix A.

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
