# Peer review of "Stereoselective [4+3]-Cycloaddition of 2-Amino-β-nitrostyrenes with Azaoxyallyl Cations to Access Functionalized 1,4-Benzodiazepin-3-ones"

_molecules, 2024, doi:10.3390/molecules29061221_

Round 1
Reviewer 1 Report
Comments and Suggestions for Authors
Based on authors’ previous work, they reported the same synthesis but used different substrates. The resulting products, 1,4-benzodiazepin-3-ones, are biologically and pharmaceutically useful compounds. [4+3]-cycloaddition was conducted between 2-amino-β-nitrostyrenes and α-bromohydroxamatein the presence of Cs2CO3 as a base. The cin-chonidine-derived squaramide catalysts led up to modest enantioselectivities and chemical yields mostly. This work is publishable in Molecules. Two points need to pay attention: 1) “high enantioselectivities” should be changed. In fact, all ee% data is modest. 2) Besides NO2 as electron-withdrawing group on the substrates, at least two more substrates containing other electron-withdrawing groups, such as SO2R, CN, etc. should be added to this work.
Author Response
Dear reviewer,
First of all, I would like to thank for your kind consideration on our manuscript.
1) This work is publishable in Molecules. Two points need to pay attention:
--> We appreciate his or her time and effort in reviewing our manuscript.
2) “high enantioselectivities” should be changed. In fact, all ee% data is modest.
--> “high” is changed to “moderate”.
3) Besides NO2 as electron-withdrawing group on the substrates, at least two more substrates containing other electron-withdrawing groups, such as SO2R, CN, etc. should be added to this work.
--> The synthesis of unprotected 2-amino-β-nitrostyrene proved challenging. Despite attempts to synthesize the CN-substituted substrate, it could not be successfully produced.
Reviewer 2 Report
Comments and Suggestions for Authors
The submitted papers describes the new synthesis of functionalized 1,4-benzodiazepin-3-one. The key reaction is [4 +3] cycloaddition reaction of 2-amino-β-nitrostyrenes with α-halohydroxamates. In fact, the reported method is an interesting deployment of previously reported [4+3] cycloaddition reactions between azaoxyallyl cation and appropriate acceptors. The research is well planned, authors described the optimization of reaction conditions in respect to applied base, solvents and in the case of enantioselective cycloaddition the catalyst. Authors successfully scaled reaction conditions to one mmol-scale synthesis. The structure of synthesized compounds is undoubtedly confirmed using spectroscopic method and structural X-ray analysis. Reading the text I have found a few questions to need comments:
1. Authors mention usage of silica 261 gel 60-F plates EM Reagents. Is it sodium cacodylate or something else? Please indicate the manufacturer.
2. For column chromatography a mixture of EtOAc and hexanes was used ? Please clarified , it was a petroleum ether boiling in a range of 61-76 oC, or mixture of hexane isomers, indicated manufacturer.
3. The purity of chiral substances is reported commonly by enantiomeric excess instead of enantiomeric ratio. Please report your data as ee.
Author Response
Dear reviewer,
First of all, I would like to thank for your kind consideration on our manuscript.
1) Reading the text I have found a few questions to need comments:
--> We appreciate his or her time and effort in reviewing our manuscript.
2) Authors mention usage of silica 261 gel 60-F plates EM Reagents. Is it sodium cacodylate or something else? Please indicate the manufacturer.
--> The information about the manufacturer (Merk) of silica gel 60-F plates was recorded.
3) For column chromatography a mixture of EtOAc and hexanes was used ? Please clarified , it was a petroleum ether boiling in a range of 61-76 oC, or mixture of hexane isomers, indicated manufacturer.
--> We used 95% n-hexane for column chromatography, commonly referred to as hexanes.
4) The purity of chiral substances is reported commonly by enantiomeric excess instead of enantiomeric ratio. Please report your data as ee.
--> The purity of chiral substances altered the enantiomeric ratio (er) to enantiomeric excess (ee).
Reviewer 3 Report
Comments and Suggestions for Authors
The authors describe an interesting extension of their previous research (Cf. refs. 20, 23 in the paper) on the (asymmetric) [4+3]-cycloadditions of azaoxyallyl cations to 2-aminophenyl enones. In this paper, the last compounds are replaced by 2-amino-beta-nitrostyrenes. Reaction conditions are similar, with minor variations: Cs2CO3 (racemic version) or Na2CO3 as a base and HFIP as co-solvent, with cinchona alkaloid-squaramide bifunctional organocatalysts.
Yields of racemic products range from moderate (41%) to good (91%). In the chiral version, a cinchonidine-derived catalyst is used (other than a quinine-derived one), yields fall generally in the 60-66% range, up to 80% in one instance). Enantiomeric purities are inferior for nitrostyrenes than for enones (72% ee maximum vs. 99% ee maximum).
New products are well-characterized, the paper is clearly written and the experimental section is very complete. Absolute configuration has been ascertained in one instance by X-ray diffraction analysis and assigned by analogy, as it usually happens, and is in agreement with that of the enone adducts. Literature references are adequate, although the citation of some general reviews on organocatalytic asymmetric cyclization reactions (Cf. Chem. Rev. 2011, 111, 4703) could be helpful to the general reader.
The proposed mechanism (Figure 4) seems plausible. Please observe that the MeO group is not present in the cinchonidine-derived catalyst III.
The hydrogenation of 3a to 5 is reported to take place with increased enantiomeric purity (from 72% ee to 80% ee), which seems strange to me since the yield is very high (94%). After inspection of the HPLC traces of 5 in the SI (page S38) my opinion is that this change in %ee is not meaningful (see a shoulder at ca. 19.0 min for the major enantiomer).
In summary, I'm convinced that this a sound an interesting work on asymmetric organocatalysis that deserves to be published in Molecules.
Author Response
Dear reviewer,
First of all, I would like to thank for your kind consideration on our manuscript.
1) New products are well-characterized, the paper is clearly written and the experimental section is very complete. Absolute configuration has been ascertained in one instance by X-ray diffraction analysis and assigned by analogy, as it usually happens, and is in agreement with that of the enone adducts.
--> We appreciate his or her compliment on our work.
2) Literature references are adequate, although the citation of some general reviews on organocatalytic asymmetric cyclization reactions (Cf. Chem. Rev. 2011, 111, 4703) could be helpful to the general reader.
--> The mentioned paper “Moyano, A.; Rios R. Chem. Rev. 2011, 111, 4703–4832” has been included as reference 20.
2) The proposed mechanism (Figure 4) seems plausible. Please observe that the MeO group is not present in the cinchonidine-derived catalyst III.
--> Checked and changed.
3) The hydrogenation of 3a to 5 is reported to take place with increased enantiomeric purity (from 72% ee to 80% ee), which seems strange to me since the yield is very high (94%). After inspection of the HPLC traces of 5 in the SI (page S38) my opinion is that this change in %ee is not meaningful (see a shoulder at ca. 19.0 min for the major enantiomer).
--> Many thanks for his or her pointing that out.
Reviewer 4 Report
Comments and Suggestions for Authors
In this manuscript, Kim and his colleagues developed a highly efficient method for synthesizing 1,4-benzodiazepin-3-ones with organocatalytic asymmetric [4+3]-cycloaddition in commendable yields and high enantioselectivities based on the previous work of the author's research group. In general, the structure of the paper is reasonable and logical, and the design principle is clearly and moderately illustrated. The obtained 1,4-benzodiazepines may provide some new insight into the compound library of bioactive drugs. However, there still are some improvements for this article. Therefore, this manuscript is recommended for publication in the molecules by addressing the following minor revisions:
1. As you mentioned in the text (Page 2, Line 66), “seven-membered 1,4-benzodiazepine-3-ones displayed promising bioactivity in the prevention of peripheral nerve degeneration” in your previous work (Adv. Synth. Catal. 2021, 363, 4197-4203). It is suggested that the author should conduct further pharmacological activity testing on newly synthesized benzodiazepine compounds to increase the novelty of the article.
2. In Scheme 2, the substituent type of substrate 1 is too homogeneous to prove the universality of the reaction. If possible, it is recommended to enrich the product variety.
3. In the scheme 1, the group in substrate 2-amino-β-nitrostyrenes is represented as R3, however in the scheme 2 and 3, the group in substrate 2-amino-β-nitrostyrenes is represented as R1. It is recommend to unify them.
4. In the screening table, some inconsistencies appeared in the tables and texts. For example, “the co-solvent system of HFIP with solvents such as CHCl3, CH3CN, toluene, and THF (Page 4, Line 101)” correspond to “Table 1, entries 5-8”, while in paper is “Table 1, entries 4-8”. Obviously this is not right.
5. It is suggested that the X-ray crystal structure of compound 3a (Figure 3) should be placed beside the corresponding compound in Scheme 3.
6. In Scheme 4, the weight of the synthesized target product in one-mmol scale synthesis could be provided, which will make the reader more intuitive.
7. Please double-check your data to make sure there are no omissions. For instance, the HRMS of compound 3k is missing (Page 13, Line 389).
8. Please see the format errors in the references. For example, in reference 32, 33 and 34, the volume number should be italicized.
Comments on the Quality of English LanguageModerate editing of English language required for this manuscript
Author Response
Dear reviewer,
First of all, I would like to thank for your kind consideration on our manuscript.
1) there still are some improvements for this article. Therefore, this manuscript is recommended for publication in the molecules by addressing the following minor revisions
--> We appreciate his or her time and effort in reviewing our manuscript.
2) As you mentioned in the text (Page 2, Line 66), “seven-membered 1,4-benzodiazepine-3-ones displayed promising bioactivity in the prevention of peripheral nerve degeneration” in your previous work (Adv. Synth. Catal. 2021, 363, 4197-4203). It is suggested that the author should conduct further pharmacological activity testing on newly synthesized benzodiazepine compounds to increase the novelty of the article.
--> We are planning activity tests, which include pharmacological activity tests for peripheral nerve degeneration, for all synthesized compounds. Additionally, activity tests on colon cancer tumors are being conducted for specific compounds.
3) In Scheme 2, the substituent type of substrate 1 is too homogeneous to prove the universality of the reaction. If possible, it is recommended to enrich the product variety.
--> We aimed to broaden the scope of substrates for the cycloaddition reaction, but encountered challenges in synthesizing diverse unprotected 2-amino-β-nitrostyrenes.
4) In the scheme 1, the group in substrate 2-amino-β-nitrostyrenes is represented as R3, however in the scheme 2 and 3, the group in substrate 2-amino-β-nitrostyrenes is represented as R1. It is recommend to unify them.
--> Checked and changed.
5) In the screening table, some inconsistencies appeared in the tables and texts. For example, “the co-solvent system of HFIP with solvents such as CHCl3, CH3CN, toluene, and THF (Page 4, Line 101)” correspond to “Table 1, entries 5-8”, while in paper is “Table 1, entries 4-8”. Obviously this is not right.
--> Checked and changed.
6) It is suggested that the X-ray crystal structure of compound 3a (Figure 3) should be placed beside the corresponding compound in Scheme 3.
--> Checked and changed.
7) In Scheme 4, the weight of the synthesized target product in one-mmol scale synthesis could be provided, which will make the reader more intuitive.
--> the weight of the synthesized target product was added.
8) Please double-check your data to make sure there are no omissions. For instance, the HRMS of compound 3k is missing (Page 13, Line 389).
--> Checked and changed.
9) Please see the format errors in the references. For example, in reference 32, 33 and 34, the volume number should be italicized.
--> Checked and changed.
Round 2
Reviewer 1 Report
Comments and Suggestions for Authors
Revision is fine